# Micromolar Dihydroartemisinin Concentrations Elicit Lipoperoxidation in *Plasmodium falciparum*-Infected Erythrocytes

**DOI:** 10.3390/antiox12071468

**Published:** 2023-07-21

**Authors:** Oleksii Skorokhod, Elena Valente, Giorgia Mandili, Daniela Ulliers, Evelin Schwarzer

**Affiliations:** 1Department of Life Sciences and Systems Biology, University of Torino, Via Accademia Albertina, 13, 10123 Torino, Italy; 2Department of Oncology, University of Torino, Via Santena 5 bis, 10126 Torino, Italy; elena.valente@unito.it (E.V.); giorgia.mandili@bioclarma.com (G.M.); daniela.ulliers@unito.it (D.U.); evelin.schwarzer@unito.it (E.S.)

**Keywords:** dihydroartemisinin, endoperoxide, ROS, 4-hydroxynonenal, *Plasmodium falciparum*, cysteine proteinase falcipain 1

## Abstract

Malaria is still the most important parasitic infectious disease. Numerous substances are known to have antimalarial activity; among them, artemisinin is the most widely used one, and artemisinin-based combination therapy (ACT) is recommended for the treatment of *Plasmodium falciparum (P.f.)* malaria. Antitumor, immunomodulatory, and other therapeutic applications of artemisinin are under extensive study. Several different mechanisms of action were proposed for dihydroartemisinin (DHA), the active metabolite of artemisinin, such as eliciting oxidative stress in target cells. The goal of this study is to monitor the generation of reactive oxygen species (ROS) and lipid peroxidation product 4-hydroxynonenal (4-HNE) by DHA in *P.f.*-infected human erythrocytes. Checking ROS and 4-HNE-protein adducts kinetics along the maturation of the parasite, we detected the highest level of 4-HNE in ring forms of *P.f.* due to DHA treatment. Low micromolar concentrations of DHA quickly induced levels of 4-HNE-adducts which are supposed to be damaging. Mass spectrometry identified the *P.f.* protein cysteine proteinase falcipain-1 as being heavily modified by 4-HNE, and plausibly, 4-HNE conjugation with vital *P.f.* proteins might contribute to DHA-elicited parasite death. In conclusion, significant 4-HNE accumulation was detectable after DHA treatment, though, at concentrations well above pharmacologically effective ranges in malaria treatment, but at concentrations described for antitumor activity. Thus, lipid peroxidation with consequent 4-HNE conjugation of functionally relevant proteins might be considered as a uniform mechanism for how DHA potentiates antimalarials’ action in ACT and controls the progression of tumors.

## 1. Introduction

To defeat malaria, humans have been looking for substances with antimalarial properties for centuries [1]. Natural products, such as quinine and artemisinin, were the first antimalarials used since antiquity [1]. More potent synthetic antimalarials are based on the molecular structures that resemble natural molecules or have completely new structures arrived from chemical databases.

Parasite development is accompanied by oxidative stress [2,3,4,5] and lipoperoxidation [6,7,8,9]. Additional oxidative stress could be deadly for parasites. This is exploited by some antimalarial compounds that are able to produce free radicals [10], whereby sometimes parasites of diverse life cycle stages differ in drug susceptibility [2,11,12].

The active form of artemisinin, DHA, was shown to exert anti-plasmodium activity with an IC50 of 1–5 nM [13,14,15], but the exact molecular mechanism of antimalarial action is still not clear and under extensive study [16,17]. Several studies have proposed several cellular targets of artemisinins with the involvement of reactive oxygen species (ROS) [16,18]. It was shown that DHA is eliciting oxidative stress in target-infected erythrocytes [15,16,19] with subsequent lipid peroxidation [20,21]. This process, in theory, could be exerted by DHA itself as endoperoxide, but it is much more efficient when exercised in synergism with the iron ion present in heme moieties [15,16,19] which continuously accumulates in the parasite, forming malarial pigment hemozoin (HZ). On the contrary, some authors deny the role of heme iron in the DHA mechanism of action [22].

After accumulation and activation inside the parasites [16], DHA strongly interferes with some crucial Plasmodium proteins in different metabolic pathways, such as unfolded protein response, protein ubiquitination, proteasome, phosphatidylinositol-3-kinase, and the eukaryotic translation initiation factor 2α [16,17,23,24]. Additionally, artemisinin was supposed to bind directly on sarcoendoplasmic reticulum calcium ATPase SERCA [25,26], suppressing the parasite growth. Thus, the antimalarial action of DHA could just partially be attributed to oxidative-related processes.

Another parasite discovered to be sensitive to DHA is *Leishmania braziliensis* with EC_50_ values of 62.3 and 8.9 µM for two different parasite stages. Moreover, DHA at a concentration of 200 µM provoked the damage of human macrophages together with the intracellular parasite. Hydrogen peroxide production and the direct effects of DHA on parasite mitochondrial bioenergetics were detected [27].

Moreover, the efficient antimalarial drug DHA was proposed as a potential drug for treating malignancies, inflammatory conditions, and other diseases as well [28,29]. Mechanistic studies of antitumor activity of DHA in HCT116 colorectal and NCI–H460 lung adenocarcinoma cells revealed lipid peroxidation in response to 50 µM DHA by BODIPY-C11 fluorescent probe and protein oxidation, which resulted in mitochondrial damage and eventually in iron-dependent cell death [30]. The anticancer effects of artemisinin were recently reviewed, and the need for further studies of the artemisinin mechanism of action was underlined [31].

The aim of our investigations was to establish the DHA concentrations capable of inducing a significant amount of ROS and the lipid peroxidation product 4-HNE to damage the plasmodium parasite and to identify possible target(s) for covalent modifications by 4-HNE.

## 2. Materials and Methods

Unless otherwise stated, all materials were obtained from Sigma-Aldrich (St. Louis, MO, USA). The final purity of compounds was above 99%.

### 2.1. In Vitro Growth of Plasmodium falciparum and Life-Stage-Specific Enrichment of Parasitized Erythrocytes

*Plasmodium falciparum* (Palo Alto strain, Mycoplasma-free)-infected erythrocytes were kept in the culture as described [8]. Fresh blood (anticoagulated by heparin) from healthy donors of both sexes was obtained from the local blood bank (AVIS, Torino, Italy). Erythrocytes were separated from plasma, white blood cells, and platelets by differential centrifugation and washed 3 times in erythrocyte wash medium (EWM, RPMI 1640 medium with the addition of HEPES), and after resuspension in EWM supplemented with 2% (*v*/*v*) SAG (150 mM NaCl/1.25 mM adenine/45 mM glucose) stored at 50% hematocrit at 4 °C for not more than 12 h. To synchronize the cultures by parasite stage, schizonts (mature forms of *Plasmodium*) were collected from the 40% to 60% interface after passing a mixed-stage culture through a discontinuous Percoll/mannitol density gradient by centrifugation at 5000× *g* for 30 min. Schizonts were added to erythrocytes resuspended at 1% hematocrit in parasite growth medium (PGM comprises EWM supplemented with 2 mM glutamine, 20 mM glucose, 0.025 mM adenine, 32 mg/mL gentamicin, and 1% (*w*/*v*) Albumax I (Invitrogen, Auckland, New Zealand)) for reinfection for 10–14 h at 37 °C under a 5% O_2_, 5% CO_2_, 90% N_2_ (*v*/*v*) atmosphere. Then, the residual mature non-reinfected parasites and expelled hemozoin were removed from the culture by centrifugation through 80% Percoll/mannitol for 30 min at 5000× *g*. The ring-stage parasites (early parasite stage) from the bottom fraction were kept for further synchronous culturing [32]. These cultures were incubated with DHA in a stage-specific manner. To obtain high parasitemia, the ring fraction of the above-described synchronous culture was additionally enriched using a Percoll/mannitol gradient (90%) at 14 h after reinfection. To enrich the trophozoites (advanced maturation stage), they were collected from the 70% to 80% interface at 30 h after reinfection. The schizonts (late maturation stage) were collected from the 40% to 60% interface after Percoll/mannitol separations of synchronous cultures performed at 48 h after reinfection. Each parasite fraction taken from the Percoll gradient was washed with EWM and maintained in PGM under standard culture conditions. Infected erythrocytes were allowed to recover for 1 h before the experimental treatments.

### 2.2. DHA and N-Acetylcysteine (NAC) Treatment of Infected and Non-Infected Erythrocytes

DHA (Selleck Chemicals GmbH, Planegg, Germany) was dissolved in water-free DMSO to obtain a stock solution of 10 mM which was further diluted with DMSO prior to use, to obtain DHA solutions of 1 mM, 100 µM, 10 µM, 5 µM, and 1 µM. The DHA solutions were added to synchronized parasite culture suspensions at a final dilution of 1:1000 (*v*/*v*) to obtain final DHA concentrations of 10 µM, 1 µM, 100 nM, 10 nM, 5 nM, and 1 nM in parasite-infected erythrocyte suspensions. The stock DHA powder was kept frozen at −80 °C, and all solutions were prepared immediately before use. Synchronous cultures with 10–13% of parasitemia were supplemented with a single dose of 0–10 µM DHA (final concentration) either at the ring stage (12–16 h postinvasion) or at the trophozoite stage (26–30 h postinvasion) and maintained under standard culture conditions (see above) until analysis. Non-infected erythrocytes were treated similarly and maintained under the same standard culture conditions. DHA was added at indicated concentrations for 6–24 h in PGM and cultures were kept at 37 °C under a gas mixture of 5% O_2_, 5% CO_2_, and 90% N_2_ (*v*/*v*). Infected and uninfected erythrocyte control cultures, which were not treated with DHA (0 µM DHA), were supplemented with a final concentration of 0.1% (*v*/*v*) DMSO, which corresponds to the DMSO concentration of DHA-treated cultures. For the assessment of the reinfection capacity of DHA-treated or untreated parasites, fresh non-infected erythrocyte suspensions in PGM (2% hematocrit) were added to synchronized parasites 42–48 h postinvasion, which have been treated or not with DHA at the ring stage, to adjust the parasitemia to approximately 5%. The parasite multiplication rate was determined by counting the ring-stage infected erythrocytes after reinfection.

To study the oxidative properties of DHA, the antioxidant thiol-reducing and radical- and 4-hydroxynonenal (4-HNE)-scavenging N-acetylcysteine (NAC) was dissolved at 100 mM in Milli-Q water and added 30 min prior to DHA supplementation into the cultures at a final concentration of 100 µM.

### 2.3. Morphological Analysis of DHA-Treated Infected Erythrocytes by Light Microscopy

Parasitemia, stage-specific parasite morphology, and morphologic anomalies of both parasite and host erythrocytes were determined in thin blood smears made at indicated times from DHA-containing or control cultures. The light microscopic examination of Diff-Quik^®^-stained (Medion Diagnostics GmbH, Düdingen, Switzerland) smear was performed using a Leica DM IRB microscope equipped with a 63× oil planar apochromatic objective with 1.32 numeric aperture and a DFC420C camera and DFC software version 3.3.1 (Leica Microsystems, Wetzlar, Germany). Parasitemia and the percentage of damaged intracellular parasites were assessed in triplicate, counting 400 to 1000 cells per smear at a time.

### 2.4. Parasite Growth and Damage Assessment by Flow Cytometry (FACS)

Parasite growth was determined by DNA quantification in infected erythrocytes by FACS at indicated time points after labeling culture aliquots with ethidium bromide (EtBr; 5 mg/mL final concentration) [33]. The fluorescence of the labeled infected erythrocytes was acquired on a FACSCalibur flow cytometer (BD Biosciences, Sunnyvale, CA, USA) and analyzed in the FL2 channel at 564–606 nm after argon laser excitation at 488 nm using the Cell Quest 3.1 software (BD Biosciences).

### 2.5. Hemoglobin Quantification

The hemoglobin concentration in the culture supernatant was assayed by heme-dependent luminol-enhanced luminescence. Luminescence was measured in a double injector luminometer (Sirius; Berthold, Bad Wildbad, Germany) as described [34,35]. All assays were performed in triplicate.

### 2.6. Quantification of Hemozoin Generation

Ring-stage infected erythrocytes (5–10 h postinfection) were enriched from synchronized cultures to >95% parasitemia using a 90% Percoll–mannitol cushion. Parasites were supplemented with 0–10 µM of DHA or kept untreated as controls under standard culture conditions for 24 h. Then, parasites (43–48 h postinfection) were hypotonically lysed and washed in excessively added lysis buffer (10 mM phosphate buffer, pH 8.0). Sedimented hemozoin was solubilized in 0.1 N NaOH/0.05% Triton X-100 (*v*/*v*) and quantified by heme-dependent luminol-enhanced luminescence [34,36].

### 2.7. Assay of ROS in Infected and Non-Infected Erythrocytes with DCF-DA and DHE by FACS

DHA-treated and untreated infected and non-infected erythrocytes were incubated at 37 °C for 3 h at 10% hematocrit in PGM containing 100 µM of the ROS probe dichlorofluorescein-diacetate (DCF-DA) [5,37] under orbital agitation. After labeling with DCF-DA, erythrocytes were washed three times with phosphate-buffered saline (PBS) supplemented with 20 mM glucose (PBS-G), and the samples were resuspended in EtBr at 10 mg/mL PGM for 15 min at 37 °C under orbital agitation for parasite DNA staining, which allows to distinguish infected from non-infected erythrocytes and the separate ROS analysis [32]. After incubation, erythrocytes were washed three times and resuspended in PBS-G. Erythrocytes were incubated with the hydrogen peroxide generating xanthine/xanthine oxidase system (X/XO) and used as a positive control for ROS production in erythrocytes [11,38]. The cultures were supplemented with X/XO (X, 1 mM; XO (EC 1.1.3.22 from bovine milk, 1–2 U/mg protein) 1 mU/mL) and incubated under the same conditions as after DHA treatment. The fluorescence of the labeled infected and non-infected erythrocytes was acquired on a FACSCalibur flow cytometer (BD Biosciences) and analyzed for DCF (the probe activated by ROS) in the FL1 channel at 515–545 nm and for EtBr in the FL2 channel at 564–606 nm after argon laser excitation at 488 nm using the Cell Quest 3.1 software (BD Biosciences). To exclude the emission overlap of the two fluorochromes, a minor electronic compensation was applied.

In separate experiments, the superoxide indicator dihydroethidium (DHE) was applied [32]. This dye exhibits blue fluorescence in the cell cytosol until it is oxidized and then links to parasite DNA switching to red fluorescence. It was added to the parasite culture at a final concentration of 25 mM for 30 min. Non-reacted DHE was washed out with PBS-G, and red emission from infected and non-infected erythrocytes was measured by FACS at 642 nm after excitation at 488 nm using a Guava flow cytometer (Merck Millipore, Billerica, MA, USA) and Guavasoft 3.1 software (Merck Millipore).

### 2.8. Assay of 4-HNE Conjugates in Infected and Non-Infected Erythrocytes by FACS

Tested infected and non-infected erythrocytes were washed with PBS-G supplemented with 1% of bovine serum albumin (PBS-G-BSA), incubated with anti-4-HNE-conjugate antibody (Abcam, Cambridge, UK) in PBS-G-BSA at 1:50 dilution for 1 h at room temperature (RT), and then washed twice with PBS-G-BSA. Bound anti-4-HNE-conjugate primary antibodies were marked by secondary FITC-conjugated F(ab) 2 goat anti-rabbit IgG [39,40]. The background signal was acquired by isotype controls. To distinguish infected cells, their percentage, and maturity, the labeling with EtBr (5 mg/mL final) was performed. The fluorescence of the infected erythrocytes was acquired on a FACSCalibur flow cytometer (BD Biosciences) and analyzed for FITC in the green channel at 515–545 nm and for EtBr in the red channel at 564–606 nm after argon laser excitation at 488 nm using Cell Quest 3.1 software (BD Biosciences). To exclude the emission overlap of the two fluorochromes, a minor electronic compensation was applied. The data were analyzed using the WinMDI 2.8 software (The Scripps Research Institute, San Diego, CA, USA) and presented as mean fluorescence intensity (MFI) [8,41].

### 2.9. Assay of 4-HNE Conjugates in Whole Protein Lysate

For protein extraction from infected and non-infected erythrocytes, cells were washed in EWM, centrifuged at 1700× *g*, sedimented, and hypo-osmotically lysed in a 10–fold excess (*v*/*v*) of ice-cold lysis buffer (10 mM K_2_HPO_4_/KH_2_PO_4_ pH 8.0 with protease inhibitor cocktail composed of 1 mM EDTA, 1 nM leupeptine, 3 mM pepstatine, 250 mM phenyl-methylsulfonyl fluoride (PMSF), sodium orthovanadate, and sodium fluoride at 1 mM, each and 100 mM Trolox) for 5 min on ice. The membranes and organelles of the parasite were sedimented at 16,200× *g* for 3 min at 4 °C, and the supernatant was discarded. The pellet was collected and washed 10 times by resuspension in fresh lysis buffer and subsequent sedimentation for 1 min at 16,200× *g*. Non-infected erythrocyte-derived preparations contained pure erythrocyte membranes, while infected erythrocyte preparations contained the remnants of the host erythrocyte membrane, as well as membranes and organelle debris from the parasite. The proteins were extracted with Laemmli buffer at 95 °C by 5 min incubation. Solubilized proteins were kept at −20 °C prior to use, and β-mercaptoethanol (5% *v*/*v*) was added to protein samples before loading to the sodium dodecyl sulphate—polyacrylamide gel electrophoresis (SDS-PAGE). Before loading, proteins were quantified by Bio-Rad Protein Assay (Bio-Rad, Hercules, CA, USA). The solubilized proteins (20 μg) were separated with an 8% acrylamide (*w*/*v*) SDS-PAGE and transferred onto a nitrocellulose membrane by Mini Trans-Blot equipment (Bio-Rad). Ponceau-S membrane staining was used to ensure equal protein loading. The membrane was blocked by PBS-BSA (PBS supplemented with 2% BSA) and incubated overnight at 4 °C with the monoclonal primary anti-4-HNE-conjugate antibody (Abcam, UK; 1:1000 dilution). Then, the membrane was washed with PBS-BSA and incubated for 1 h with the secondary HRP-conjugated anti-mouse antibody (Pierce, Walhem, MA, USA) at 1:10,000 dilution [42]. Positive bands were visualized by ECL and digitalized with ChemiDoc MP Imaging System and ImageLab 6.1 software (both Bio-Rad). 

### 2.10. Identification of 4-HNE-Modified Proteins and 4-HNE Binding Sites by Mass Spectrometry

Identification of trophozoite cysteine proteinase (falcipain 1) and its 4-HNE binding sites by mass spectrometry was based on the method previously reported [32,41,43]. Proteins were extracted from infected and non-infected erythrocytes by the same procedure described in the above section “Assay of 4-HNE conjugates in whole protein lysate”. The 20 μg of proteins were run in a 10% polyacrylamide gel SDS-PAGE. Then, the gel was stained with colloidal blue Coomassie (18% *v*/*v* ethanol, 15% *w*/*v* ammonium sulphate, 2% *v*/*v* phosphoric acid, 0.2% *w*/*v* Coomassie G-250) at RT for 48 h. Gel slices with the protein bands from the stained gels were excised as thin as possible (including very weak bands) and transferred into separate tubes, then unstained by five washes in 50% *v*/*v* acetonitrile in 5 mM ammonium bicarbonate (AmmB), and dried using pure acetonitrile. The gel slices were rehydrated for 45 min at 4 °C in 20 μL of AmmB containing 10 ng/μL of trypsin. Excess trypsin solution was removed, and the volume was adjusted using AmmB to cover the gel slices. Trypsin digestion was then continued at 37 °C overnight. Samples were loaded onto the MALDI target support using 1.5 μL of the tryptic protein digest mixed 1:1 with a solution of α-cyanohydroxycinnamic acid (10 mg/mL) in 60% *v*/*v* of 0.1% trifluoroacetic acid and 40% *v*/*v* of acetonitrile. Mass spectrometry (MS) analysis was performed on a MALDImicroMX spectrometer (Waters, Milford, MA, USA), equipped with a delayed extraction unit, according to the tuning procedures indicated by the manufacturer, working in reflectron mode.

Protein Lynx Global Server (Waters) was used to generate the peak lists. The 25 most intense mass peaks were applied for database searches against the NCBI and Swiss-Prot (UniProt) databases using the search program Mascot (Matrix Science, http://www.matrixscience.com (accessed on 3 May 2023)), introducing (i) 4-HNE conjugates with lysine (K), cysteine (C), and histidine (H) and (ii) methionine (M) oxidation as a variable modification from the default Mascot list of modifications. Further search parameters included taxa *Plasmodium falciparum*, trypsin digest, monoisotopic peptide masses, protein molecular mass, two eventual missed cleavages by trypsin, and peptide mass deviation tolerance of 0.5 Da. Identification of protein bands was obtained from the triplicate analysis. The proteins, considered in this study, had a Mascot score higher than 39 for SwissProt and higher than 67 in NCBI searches, suggested by Mascot to be significant.

### 2.11. Statistical Analysis

The values from at least three independent biological replicates are presented in histograms (means ± SE). The statistical significance of the difference between the groups’ means was calculated by the non-parametric Mann–Whitney U test (PASW Statistics 18, IBM SPSS, Armonk, NY, USA). If not otherwise indicated, *p* values below 0.05 were considered statistically significant. 

## 3. Results

### 3.1. DHA Affects Ring-Stage P.f. Development and Impairs Reinfection

We observed the strong detrimental action of DHA on ring-stage parasites (Figure 1 and Figure 2). This coincides with the beginning of hemozoin formation from hemoglobin, which the parasites capture from host erythrocytes. We observed the first signs of parasite damage by 10 nM DHA treatment after 3 h. The progress in damage and death after 6–24 h (Figure 1B) confirmed the low nanomolar DHA concentrations as functional against the parasites. The stained smears in Figure 1B show (i) damaged and shrank parasite forms and (ii) strongly delayed development of normally shaped parasites after DHA treatment with 10 nM and higher concentrations (Figure 1B and parasitemia count, shown below), while parasites grew regularly in untreated erythrocytes from the same donor cultured in parallel under same conditions, differentiating to late ring/early trophozoite forms after 6 h and to schizonts after 24 h from the start of observation and DHA treatment (T = 0, ring forms) (Figure 1B).

Ethidium bromide staining of parasites’ nucleic acids showed a strong delay in parasite growth. Nucleus DNA content was significantly lower (−45 ± 9%) at 6 h after treatment of ring-form parasites with 5 nM of DHA as compared to untreated parasites, and it moderately decreased further at concentrations up to 10 µM DHA (Figure 2). Due to the binding of ethidium bromide to damaged and dying parasites, the fluorescence level remained above 0 even at the highest tested DHA concentrations.

A significant slowing of parasite growth was also observed when trophozoite-stage infected erythrocytes (26–30 h postinvasion) were treated with DHA. As seen with rings, ethidium binding was significantly decreased by 5 nM DHA (−33 ± 10%) and higher concentrations as compared to untreated cultures (Figure 2, black columns).

To discriminate between damaged and delayed but regular-shaped parasites under DHA treatment, damaged parasite forms were counted separately from total parasitemia (Figure 3A). After 10 nM of DHA treatment, the percentage of parasites with irregular shapes, such as distorted, broken, pyknotic, and fragmented forms, was 7.4 ± 0.9% in the case of a total parasitemia of 11.2 ± 1.3%. Thus, a substantial 66.1% of counted parasites were visibly damaged after treatment with 10 nM DHA for 24 h. The percentages of damaged parasites increased even to 89 and 90% at 100 and 1000 nM of DHA, extremely high DHA concentrations for the herein tested sensitive *P.f*. strain. We also observed dead extracellular parasites likely expelled from erythrocytes, explaining the decrease in total parasitemia in treated cultures (Figure 3A). Loss of infected erythrocytes by DHA-induced cell lysis was excluded by sensitive hemoglobin measurement in the culture supernatants (Figure 3C). Hemozoin production by parasites was regular in untreated infected cells, increasing from early trophozoite forms to schizonts, detected microscopically (Figure 1B) and by luminescence assay. After DHA treatment with 10–100 nM, the hemozoin formation was drastically decreased.

Additionally, DHA strongly impaired the reinfection of RBC by parasites by 76% at low and by 96–97% at high nM concentrations (Figure 3B). The multiplication rate in untreated control cultures was 3.8 ± 0.4, while DHA added at 10 nM to ring-stage parasite cultures decreased the multiplication rate to just 0.9 ± 0.4 during the observation interval of 9 h. At higher DHA concentrations, the reinfection was nearly abolished, and the multiplication rates were 0.15 ± 0.1 and 0.1 ± 0.8 with 100 and 1000 nM DHA, respectively.

### 3.2. DHA Generates Free Radicals in Infected Erythrocytes

The DHA-elicited free radicals’ production in infected RBCs was assessed through DCF fluorescence after 3 h of DHA treatment on ring-stage parasite cultures and achieved 35 ± 14% and 65 ± 18% of fluorescence maximum at 1 and 10 µM, respectively. The fluorescence maximum was assessed in RBCs of the respective donor kept in DHA-free parasite cultures and incubated with the potent ROS-producing xanthine/xanthine oxidase system (Figure 4). Lower than micromolar DHA concentrations did not provoke additional free radical production in infected RBCs and achieved approximately 26% of the fluorescence maximum, as observed in DHA-free infected erythrocytes. The radical scavenger NAC significantly decreased the fluorescence elicited by 10 µM DHA (the maximal tested dose) by 38.5% (Figure 4). DHA did not elicit free radical production in non-infected RBCs at any concentration tested, and DCF fluorescence achieved 5% of the fluorescence maximum in both DHA-free control or DHA-treated erythrocytes.

Similarly, the superoxide probe DHE revealed significant ROS formation by micromolar DHA concentrations in infected erythrocytes (Figure 4B). DHA at 1 and 10 µM provoked a 1.68 ± 0.16- and 2.75 ± 0.91-fold increase in DHE fluorescence signal, respectively, in infected erythrocytes, compared to untreated infected cells. NAC was able to decrease the DHE signal elicited by 10 µM DHA by 41.2 ± 14.5%. Non-infected erythrocytes were negative for the DHE probe by themselves and after DHA treatment. The X/XO positive control ROS-eliciting system was able to induce maximum signal, measured both by DCF (Figure 4A) and DHE (Figure 4B).

### 3.3. DHA Induces Lipoperoxidation and 4-HNE-Protein Conjugation in Infected Erythrocytes

Accumulation of protein conjugates with 4-HNE, the final product of ROS formation and lipoperoxidation, was observed in DHA-treated infected erythrocytes. Untreated ring-infected erythrocytes showed a slow progressive increase in 4-HNE conjugate formation during 24 h culture, when rings became trophozoites and schizonts (Figure 5A, white columns) consistent with previous results [32,44]. In DHA-treated cultures, the progressive time- and dose-dependent increase in 4-HNE conjugates in infected erythrocytes was stronger and the conjugate level exceeded untreated cultures at DHA concentrations of 1 µM already after 3 h (+36 ± 6%). After 24 h, 100 nM and 1 µM DHA provoked increased 4-HNE conjugations by 92 ± 5 and 99 ± 4%, respectively, compared to untreated parasites (Figure 5A), indicating the accumulation of lipoperoxidation products over time, with probable harmful effects on modified proteins of growing ring-, trophozoite-, and schizont forms.

In non-infected erythrocytes from the same cultures, 4-HNE conjugation was low and DHA-independent during 24 h observation (Figure 5B). The lack of effect of DHA on non-infected erythrocytes in the culture (Figure 5B) can be explained by the necessity of parasite DHA-activating factors for the strong 4-HNE production observed here (Figure 5A). The non-significant minor increase in 4-HNE conjugation from time 0 to 24 h could be caused by 4-HNE, which diffuses out of infected erythrocytes and binds to the membrane of adjacent non-infected erythrocytes. Incubation of non-infected erythrocytes with 1–1000 nM of DHA in the absence of parasitized erythrocytes did not result in the appearance of 4-HNE conjugates. Consistent with previous observations [44], 4-HNE conjugates in trophozoites exceeded those in ring forms at T = 0 and continued to increase during a further 16 h of parasite growth to schizont forms (Figure 5C, white columns). The high levels of 4-HNE conjugates in trophozoites were further increased by DHA treatment, with peak levels reached after 16 h (Figure 5C), but differences to the parasite-elicited 4-HNE conjugates were insignificant. Note that a significant increase in 4-HNE-conjugates’ level due to DHA action was observed only in rings (Figure 5A,C), probably because the antioxidant system of the parasite was still not fully expressed in the ring stage and was not able to counteract ROS and 4-HNE produced by DHA.

Non-infected erythrocytes in trophozoite-stage cultures showed higher basic levels of 4-HNE than non-infected erythrocytes from ring-stage cultures (Figure 5D, white columns), confirming increased 4-HNE production by more mature parasites and an increased spread of 4-HNE from infected erythrocytes to non-infected adjacent cells. DHA treatment was not able to increase the level of 4-HNE-conjugates (Figure 5).

### 3.4. Protein Targets of 4-HNE and Their Potential Functional Impact

The increase in 4-HNE surface adducts is accompanied by a series of protein modifications detectable in the cell lysate by WB. Untreated ring-stage infected erythrocytes showed few modified proteins with low modification levels, as judged from WB analysis. The load of 4-HNE-modifications was increased in trophozoite-stage infected erythrocytes, as previously shown by us [32,44]. DHA treatment of infected erythrocytes at concentrations of 1–100 µM additionally increased the protein load with 4-HNE adducts and prompted us to search for specific proteins modified by 4-HNE due to DHA treatment. We performed mass spectrometry (MS) analysis after 1 µM DHA treatment for 6 h (conditions, where 4-HNE-adducts were detectable by FACS/WB and morphological changes in parasites were evident). MS analysis individuated the protein cysteine proteinase falcipain 1 as a heavily modified protein after treatment of ring-infected erythrocytes with 1 µM DHA. Without DHA, no 4-HNE modification was detectable in falcipain-1 of control parasites cultured in parallel. The mass spectrometric analysis will reveal further protein targets for 4-HNE. Falcipain-1 carried adducts and Schiff bases formed with 4-HNE (Table 1). Ten lysine residues, two histidine residues, and one cysteine residue were modified by 4-HNE, and five oxidative methionine modifications were detected without any known consequence for parasite development.

We performed BLASTp 2.12.0+ protein analysis in UniProtKB database to understand the potential consequences of conjugations. The N-terminal segment (amino acid (aa) 1–230) of *P.f.* cysteine proteinase falcipain-1 had no similarities with other *P.f.* falcipains or other cysteine proteases. The N-terminal segments differ in length and are usually shorter in other proteases than in falcipain-1. In contrast, the active-site harboring region (aa 230–569) resembled structurally *P.f.* falcipain-2 (UniProt Q56CY9) with 35% identities, 57% positives (indicating conservative replacement of amino acids), and 8% gaps in the amino acid sequence and *P.f.* falcipain-3 (UniProt Q8IIL0) with 32.6% identities, 60% positives, and 9% gaps. Similarly, numerous human cathepsins, such as cathepsin K (CATK, UniProt P43235), showed 32% identities, 47% positives, and 16% gaps in primary structure; cathepsin B (CATB, UniProt P07858) showed 23% identities, 39% positives, and 27% gaps; cathepsin S (CATS, UniProt P25774) showed 31% identities, 46% positives, and 17% gaps, compared to *P.f.* falcipain-1.

Six of the thirteen detected 4-HNE conjugation sites within *P.f.* falcipain-1 were in the N-terminal aa 1–230 region and seven in the aa 230–569 region. Comparative sequence analysis of selected acidic cysteine proteases (BLAST) revealed that the amino acids at sites corresponding to the latter seven 4-HNE modification sites in falcipain-1 were conserved in *P.f.* falcipains-2 and 3 (Table 1, right column). Both, falcipains-2 and 3 are crucially involved in parasite development. Similarly, human cathepsins have identical residues as falcipain-1 at corresponding sites. Thus, falcipain-1 cysteine C388 corresponds to C170 of human cathepsin K. C170 is reported to form a disulfide bond with C210 (https://www.uniprot.org/uniprotkb/P43235, accessed on 22 June 2023) and its conjugation with 4-HNE would structurally affect cathepsin K (Table 1, right column). Similarly, falcipain-1 cysteine C388 corresponds to C141 from cathepsin B, participating in disulfide bond C141-C207 (https://www.uniprot.org/uniprotkb/P07858, accessed on 13 July 2023) and to C170 from cathepsin S, forming disulfide bond C170-C213 (https://www.uniprot.org/uniprotkb/P25774, accessed on 13 July 2023).

## 4. Discussion

DHA was confirmed herein to efficiently damage middle-aged ring forms at low nanomolar concentrations. Morphological changes in parasites kept in synchronized cultures occurred as early as 3 h after drug exposure at 10 nM (Figure 1), and the majority of parasites were damaged during the first erythrocytic cycle (Figure 3). The few residual alive parasites re-infected reluctantly. DHA concentrations starting from 100 nM eliminated vital parasites and impaired re-infection almost completely.

Several partially contrasting studies were published on the molecular mechanism behind DHA efficiency against *P.f.* [16,19,22,45], which also included the hypothesis of lipid peroxidation as the cause of parasite damage [15]. The latter one seemed to be most interesting for two reasons. Firstly, the reported successful application of probucol in combination with DHA as antimalarial treatment in mice increased DHA efficiency in terms of parasitemia control and host survival rate [46]. Probucol eliminates alpha-tocopherol, the potent lipophilic antioxidant which protects membranes from lipid peroxidation. Secondly, lipoperoxides give rise to a series of biologically active compounds such as 4-HNE, which is able to bind covalently to biomolecules altering their function which might be crucial for a protozoan as *P.f*.

Here, we show that DHA was able to elicit radicals and ROS in *P.f.*-infected erythrocytes. However, concentrations of DHA as high as 1 µM were necessary to obtain a measurable signal from intact *P.f.*-infected erythrocytes with the radical-sensitive fluorescence probe DCF in flow cytometry. The need for relatively high DHA concentrations might be due to highly efficient radical trapping in the parasite–host system. DHA is enriched in the parasite compartment, which means it passes through at least three, likely four, lipid bilayers, i.e., plasma membrane of the host cell, parasitophorous vacuole membrane, plasma membrane, and eventually the membrane of the digestive vacuole of the parasite to finally be detectable in neutral lipid bodies, most likely in the digestive vacuole of the parasite [16]. Just here, the activation of DHA may occur when heme iron from hemoglobin digestion cuts the endoperoxide bridge to activate the molecule and where the radical cascade has its origin.

A more sensitive approach according to us was monitoring lipid peroxidation by its final product, 4-HNE, which can pass cell membranes thanks to its amphiphilic character, is highly reactive, and binds stably to biomolecules. 4-HNE binds to the accessible amino acid residues of histidine, cysteine, and lysine in proteins. Formed Schiff bases and Michael adducts are stable, and 4-HNE accumulates over time, offering sensitive detection of ongoing lipid peroxidation. The study shows that DHA provoked lipid peroxidation in middle-aged ring stages very quickly. At 3 h, a dose-dependent increase in 4-HNE conjugates in the membrane of the infected RBCs occurred at 5 nM and higher, though significant differences to untreated parasites were seen only at 1 µM. We have no data yet on whether proteins inside the cell or even the parasite are modified at lower overall doses of DHA. During long-term exposure of cells, DHA-elicited lipoperoxidation increased dose-dependently, becoming significantly different from untreated control cultures at 100 nM and higher. Thus, lipoperoxidation products elicited by DHA were detectable at 10 times higher concentrations as needed for parasite death. For sure, host RBC membrane protein–HNE conjugation, as seen in vitro at 100 nM DHA, may change the cellular characteristics of the host cell, such as rigidity, making the cell vulnerable to phagocytosis, as shown in in vitro studies with 4-HNE [8,44].

Similar DHA concentrations (100 nM) were shown to change the mitochondrial membrane potential of Plasmodium, and at 1 µM, the components of parasite electron transport complex were inhibited by 5–21% [15], which could be signs of structural damage caused by lipid peroxidation.

In this study, 4-HNE formation and conjugation with proteins were provoked by micromolar DHA concentrations, at which DHA exerts antitumor activity [28,29]. The molecular mechanism of DHA action against tumors is largely unknown, and we propose 4-HNE binding to functionally relevant proteins as the molecular mechanism through which DHA might control tumor progression. In light of the conflicting data in the literature on the role of 4-HNE in tumor genesis, attributing 4-HNE a second messenger role [47], or observing excessive 4-HNE production with modifications of proteins, lipids, and DNA during cancer pathogenesis [48], the finding of the causal role of cathepsins in tumor progression is very important [49]. In our study, we found the cysteine-protease falcipain-1 heavily modified in the parasite. Interestingly, in search for structural similarities with other functionally similar proteases, such as cathepsins, sequence comparison revealed the relative conserved structure in the active-site harboring region of human cathepsin K, including several of the 4-HNE-binding amino acids found in falcipain-1.

This is quite exciting as 4-HNE, a nine-carbon atom-long, binds covalently to accessible amino acid residues and introduces its rather long hydrophobic tail in the protein structure. Occurring near the active site, this should have functional consequences and inhibit the protease activity. Cathepsins together with other proteases help to form premalignant lesions and a favorable local microenvironment for tumor cells through the modulation of growth factors, chemokines, cytokines, and their receptors [49]. Thus, it is not surprising that they were proposed as a promising target for new antitumor and immune therapies [49,50]. Due to the different tissue localization of cathepsins and different modes of potential involvement in tumor genesis, each cathepsin needs to be studied individually as a potential target for DHA action via 4-HNE-conjugate formation. The falcipain-1-similar cathepsin K is overexpressed in prostate and breast cancer [49] and might be a likely and promising target for 4-HNE due to DHA treatment.

Also, lipid peroxidation as a consequence of DHA treatment is the trigger for subsequent ferroptosis, which underlies the relatively modern strategy to fight cancer cell growth. In cancer cells with high iron load, high DHA concentrations above 10 µM provoke sufficient lipid peroxidation for desired growth inhibition by DHA [51].

## 5. Conclusions

In summary, cathepsin modification by 4-HNE could be the new mechanism both for DHA action and for pathologies associated with strong lipoperoxidation, such as chronic inflammations or age-related diseases.

Having seen 4-HNE conjugates with proteins on the host cell surface, rather distant from the likely origin of DHA-elicited peroxidation in the intracellular parasite compartment, it is not surprising that intracellular and parasite proteins are modified by the lipid peroxidation product 4-HNE due to DHA treatment. Here, we report for the first time a parasite protein, the cysteine protease falcipain-1, to be heavily modified by 4-HNE in DHA-treated parasites. Along the 569-amino acid-long protein chain, thirteen 4-HNE molecules are covalently bound to amino acid residues which are accessible for the 9-carbon-long amphiphilic 4-HNE. Each 4-HNE molecule introduces an aliphatic hydrophobic moiety, which plausibly has consequences for the protein structure and also the enzyme activity as seven of 4-HNE binding sites were localized in the active-site harboring region. Functional losses by 4-HNE binding were demonstrated with several enzymes and receptors [41,52,53]. Falcipain-1 has an acidic pH optimum at 5–6.5, which could indicate its localization in the food vacuole. Although discovered as the first in the falcipain family, others, falcipains 2, 2′, and 3, were meanwhile functionally better characterized in their role to digest hemoglobin [54,55]. Thus, the functional consequences of the plentiful alkylation along the whole falcipain sequence for the parasite life cycle with the nine-carbon atom chains of 4-HNE remain concealed, yet. Not unlikely it may impact merozoite invasion into host erythrocytes [55,56].

Sequence comparison revealed similarities between falcipain-1 and falcipains-2 and 3 in the active-site harboring region, aa 230–569. It may be meaningful that several of the amino acid positions that were conjugated with 4-HNE-in falcipain-1 are conserved in falcipains-2 and 3 and can be considered as potential 4-HNE targets. In contrast to falcipain-1, both other proteases are crucially important for parasite survival. Thus, the final decision on whether 4-HNE has a role in DHA action or not needs the unbiased analysis of the whole parasite proteome for 4-HNE modification at low nanomolar DHA concentration, an analysis that was not performed yet.

## Figures and Tables

**Figure 1 antioxidants-12-01468-f001:**
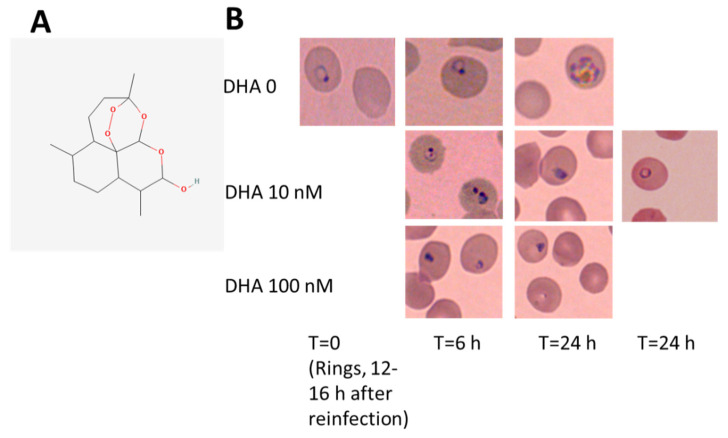
**Common DHA effects on ring-stage *P.f.*-infected RBC.** (**A**) The chemical structure of DHA is shown with oxygen atoms in red, including the functional endoperoxide group (National Center for Biotechnology Information. PubChem Compound Summary for CID 540327, Dihydroartemisinin, https://pubchem.ncbi.nlm.nih.gov/compound/Dihydroartemisinin, accessed on 30 May 2023). (**B**) Morphological damage and growth delay of *P.f*. parasites were assessed in synchronous in vitro cultures of asexual stages of parasites grown in erythrocytes and treated at the ring stage (time 0) with 0, 10, and 100 nM DHA for 6 and 24 h. Microscopic images taken after Diff-Quik^®^-stained smears show damaged and shrunken parasite forms after DHA treatment. Images were acquired using an inverted microscope Leica DM IRB (Leica Microsystems, Wetzlar, Germany) equipped with a 100× oil planar apochromatic objective with 1.32 numerical aperture and a Leica camera DFC 420 C. Leica DFC software (version 3.3.1) was applied.

**Figure 2 antioxidants-12-01468-f002:**
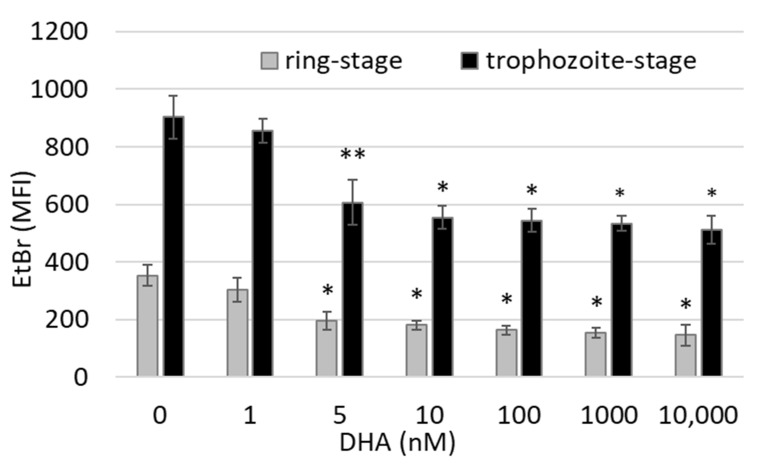
**Inhibition of parasite growth by DHA**. Synchronized *P. falciparum* cultures were treated at the ring stage (12–16 h after reinfection, grey columns) and the trophozoite stage (26–30 h after reinfection, black columns) with 0–10,000 nM DHA. Parasite growth was assessed at 6 h after the start of DHA exposure by flow cytometry after cell staining with ethidium bromide (EtBr). Mean fluorescence intensities (MFIs) are shown as mean ± SE of 3–5 independent experiments performed with erythrocytes of different donors. The significance of differences vs. untreated control cultures (0 µM DHA) is indicated by * for *p* < 0.01 and ** for *p* < 0.05.

**Figure 3 antioxidants-12-01468-f003:**
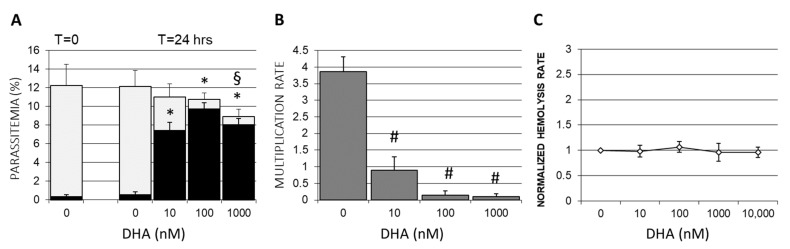
**Damage of ring-stage P. falciparum and decrease in multiplication rate by DHA**. (**A**) Synchronized *P. falciparum* cultures were treated at the ring stage (12–16 h after reinfection) with 0–1000 nM DHA (time 0; T = 0) and examined for parasitemia at 24 h after DHA supplementation (T = 24) by manual microscopic inspection of Diff-Quik^®^-stained smears. Parasites with irregular shapes (distorted, broken, pyknotic, and fragmented forms) were counted as damaged (black columns) separately from infected erythrocytes harboring normally shaped viable parasites (gray columns). (**B**) The multiplication rate was determined for *P.f.*, which were treated at the ring stage with DHA and calculated as the ratio of parasitemias of viable parasites after and before reinfection of freshly added erythrocytes. The parasitemia before reinfection was adjusted by supplementation with fresh non-infected erythrocytes at a schizont parasitemia of 4.93 ± 0.6% in the case of not DHA-treated synchronous control cultures (DHA 0 nM). Parasite cultures pre-treated with 1–1000 nM DHA at the ring stage were identically supplemented with fresh erythrocytes prior to reinfection, even though alive schizont parasitemias were lower due to DHA treatment as compared to untreated control culture. Parasitemia after reinfection was counted 9 h after adding non-infected donor erythrocyte suspension. (**C**) The absence of DHA-elicited hemolysis is shown as a normalized hemolysis rate. Free hemoglobin was quantified in culture supernatant after 6 h of DHA treatment. Supernatant hemoglobin was referred to as total hemoglobin in the RBC suspension to obtain the hemolysis rate. These rates were normalized by referring any measured value to the hemolysis rate assessed in respective untreated cultures with erythrocytes from the same donor. Means ± SE are shown for independent cultures with RBCs from 3 different donors. The significances of differences between treated (0–1000 nM DHA) and untreated control cultures (0 nM DHA) are indicated by * for *p* < 0.05 for damaged parasites, § for *p* < 0.05 for total parasitemia (viable plus damaged parasites), and # for *p* < 0.05 for multiplication rate of parasites.

**Figure 4 antioxidants-12-01468-f004:**
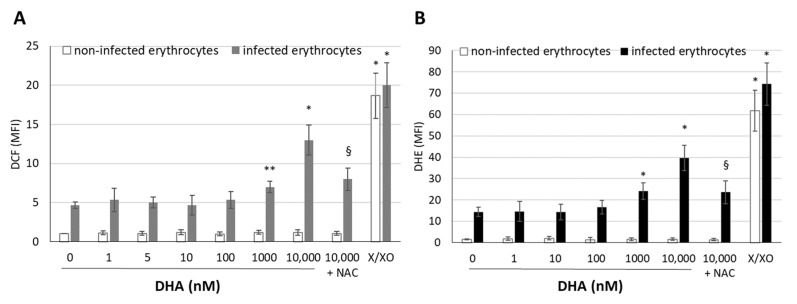
**ROS production by DHA in infected erythrocytes**. Synchronized *P. falciparum* cultures were adjusted to 10% parasitemia and treated with 0–10 µM DHA at the ring stage (12–16 h after reinfection). Where present, 100 µM NAC was added 30 min prior to DHA to the parasites culture. In parallel, an otherwise untreated culture was incubated with pro-oxidant xanthine/xanthine oxidase as a positive control for ROS production. Cells were stained with the ROS-sensitive fluorescent probes DCF-DA (**A**) and DHE (**B**) at 3 h after pre-incubation with DHA. The probe was separately assessed in non-infected (white columns) and infected (grey/black columns) erythrocytes distinguished by ethidium bromide using flow cytometry. The mean fluorescence intensity (MFI) of labeled cells was normalized by setting the MFI of corresponding non-infected, not-DHA-treated erythrocytes as 1. Means ± SE for 3–5 erythrocyte donors are plotted. The significance of the differences vs. untreated control cultures (0 µM DHA) is indicated by * for *p* < 0.01 and ** for *p* < 0.05. The significance of difference between 10 µM DHA-treated *P.f.* cultures pretreated with NAC or not is indicated by §.

**Figure 5 antioxidants-12-01468-f005:**
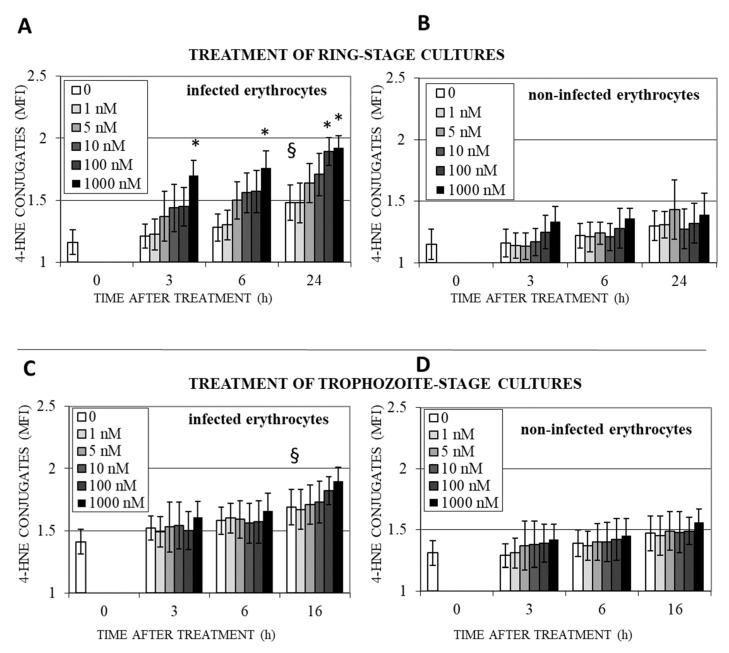
**Slow increase in 4-HNE conjugate formation in infected erythrocytes by DHA**. Synchronized *P.f.* cultures were treated at the ring stage (12–16 h after reinfection, **A**,**B**) and the trophozoite stage (26–30 h after reinfection, **C**,**D**) with 0–1000 nM DHA for 3–24 h. 4-HNE-conjugate formation was assessed in intact infected (**A**,**C**) and non-infected erythrocytes from the same cultures (**B**,**D**) using flow cytometry at indicated time points after DHA addition at time 0 and expressed as MFI normalized to isotype control. Means ± SE are presented for independent cultures from 4 RBC donors (**A**,**B**) and 3 RBC donors (**C**,**D**). The significance of differences between DHA-treated and non-treated cultures (**A**–**D**) at the same time point is indicated by * for *p* < 0.05. The significance of differences between untreated cultures (concentration 0, white columns) at different time points and the starting point of observation (time 0) is indicated by § for *p* < 0.05.

**Table 1 antioxidants-12-01468-t001:** DHA induced protein modification of cysteine proteinase falcipain-1 (Uniprot: Q8I6V0_PLAF7, https://www.uniprot.org/uniprotkb/Q8I6V0 (accessed on 30 May 2023) and PlasmoDB: PF3D7_1458000, https://plasmodb.org/plasmo/app/record/gene/PF3D7_1458000 (accessed on 25 May 2023)). Michael adducts formed between 4-HNE and lysine (K), histidine (H), or cysteine (C) residues are in italics, and Schiff bases formed between 4-HNE and lysine residues are in bold. Underlined amino acid pairs are alternatively modified by 4-HNE. Methionine oxidation sites are reported in column Oxidation. Aa residues in selected proteases that correspond to 4-HNE modification sites in falcipain-1 are listed in the column on the right. Corresponding modification sites in falcipain-1 are presented in parenthesis.

4-HNE Binding Site (UniProt)	Oxidation (UniProt)	Stage-Dependent Expression/Assessment	Known Molecular Function (UniProt)	Amino Acid Residues in Cysteine Proteases Corresponding to 4-HNE Binding Sites in Falcipain-1
**K88**, *K126*, **K147**, **K209**, **K211**, **K224/K227**, *K288*/*K300*, *H292*, *H296*, **K375/K390**, *C388*, *K436*, *K446*	M204, M208, M243, M297, M545	Whole parasite cycle including blood stage/transcriptome	Cysteine-type peptidase activity	*P.f.* falcipain-2: C316 (*C388*), K365 (*K436*); *P.f*. falcipain 3: K105 (**K147**), H238 (*H292*), K311 (**K375**), C324 (*C388*); Human cathepsin K: H99 (*H296*), C170 (*C388*); Human cathepsin B: C141 (*C388*); Human cathepsin S: C170 (*C388*);

## Data Availability

Data are comprised within the article.

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
