# Peer review of "Micromolar Dihydroartemisinin Concentrations Elicit Lipoperoxidation in Plasmodium falciparum-Infected Erythrocytes"

_antioxidants, 2023, doi:10.3390/antiox12071468_

Round 1
Reviewer 1 Report
This paper takes a detailed look at the possible mechanisms of artesunate (and its active ingredient dihydroartemisinin (DHA)) on P. falciparum parasites. The group concentrates on the possible oxidative damage caused by these compounds. The experiments are well performed and appropriately analyzed. The manuscript suffers from the (acknowledged) fact that most effects of DHA are seen are at levels 100-1000X greater than the DHA levels where antimalarial activity is seen. The authors have various explanations for this this, including the accessibility of some of the measured products given the complex membrane structure of the parasite. These explanations are plausible – but do not mask the data which shows minimal oxidant effect (as measured by both ROS and 4-HNE) at drug levels where DHA has antimalarial activity. The one Figure where DHA shows effect at nM levels (Figure 5A) only results in 4-HNE levels that are seen later in the malaria life cycle in untreated cells.
Specific areas of concern:
Line 267-269 claims that DHA treatment delays the normal development of the parasite (and references Figure 1B). I do not see evidence of normal parasites (even at later time points in any of the DHA treated cultures). Is there truly evidence for a delay in maturation or is it death with no progression? Evidence for a delay is not shown.
The authors point out the problems of using EtBr staining as a measure of parasite viability, given the staining of free DNA from dead parasites. Given that the read out in Figure 2 does not closely match the microscopic data in Figure 1, the use of this metric seems questionable.
The authors mention in the abstract and the introduction that the artemisinin class of compounds may have possible antitumor and immunomodulatory effects in addition to their well-known antimalarial effects. These possible ‘repurposed’ therapies are not further explored in the Discussion section. Given the lack of alterations at concentrations where anti-malarial effects take place, perhaps the authors should spend more time discussing these possible non-malarial applications.
A broader analysis of protein modifications needs to be undertaken prior to publication. The reader is left questioning whether this modifications of falcipain-1 are novel to this particular protein (ie only 1% of parasite proteins are modified) or universal (ie 99% of parasite proteins are modified). It is unclear what concentration of DHA was used in the mass spectroscopy experiments. It is also unclear how the authors found this specific drug-protein interaction – whether it was via a directed or an unbiased approach.
Falcipain – 1 is a non-essential protein in P. falciparum; parasites without this gene function normally in respects to growth and invasion (PMID: 15166288). It is therefore difficult to imagine that DHA modified falcipain-1 could explain parasite death. The data presented do not even comment on the functionality of the enzyme with these modifications. Is there evidence that it has decreased function when modified in the ways described?
The morphologic and parasite death data that is in the first paragraph of the Discussion is not novel, and I am not sure should even be covered.
Given that ROS and 4-HNE effects are only seen at very high drug concentration, the physiologic relevance needs to be considered. There is some mention in Abstract (Line 12) and Intro (Line 61-68) but nothing in discussion. This is not a paper about the pharmacologic mechanisms of Artesunate in parasite infected cells – as these effects do NOT take place at the nM levels where DHA has its effect. The adduct information on falcipain-1 is interesting but again needs to be put into context. How do the authors hypothesize this to influence parasite death if only seen at uM range?
There are several instances where English grammar and diction could be improved.
Author Response
Reviewer 1
This paper takes a detailed look at the possible mechanisms of
artesunate (and its active ingredient dihydroartemisinin (DHA)) on P.
falciparum parasites. The group concentrates on the possible oxidative
damage caused by these compounds. The experiments are well performed and
appropriately analyzed. The manuscript suffers from the (acknowledged)
fact that most effects of DHA are seen are at levels 100-1000X greater
than the DHA levels where antimalarial activity is seen. The authors
have various explanations for this this, including the accessibility of
some of the measured products given the complex membrane structure of
the parasite. These explanations are plausible – but do not mask the
data which shows minimal oxidant effect (as measured by both ROS and
4-HNE) at drug levels where DHA has antimalarial activity. The one
Figure where DHA shows effect at nM levels (Figure 5A) only results in
4-HNE levels that are seen later in the malaria life cycle in untreated
cells.
Specific areas of concern:
Line 267-269 claims that DHA treatment delays the normal development of
the parasite (and references Figure 1B). I do not see evidence of normal
parasites (even at later time points in any of the DHA treated
cultures). Is there truly evidence for a delay in maturation or is it
death with no progression? Evidence for a delay is not shown.
Reply: We agree, Figure 1 was intended to present those damaged parasite morphologies that were most frequently observed in the synchronized in vitro cultures under DHA-treatment to demonstrate that the drug is functional in our tests. Nonetheless, a certain percentage of parasites survived a DHA treatment at 12-16 h post-invasion without clear signs of damage after 24 h incubation with DHA, but with a heavy delay in development to trophozoite and schizont stages. These parasites remained morphologically rings while DHA-free control cultures kept in parallel reached already the schizont stage. To illustrate the survival state of parasites under DHA we add an example of a survived 36-40 h old parasite still with ring morphology from a culture that was exposed to 10nM DHA at 12-16 h post-invasion, into Fig 1B.
The authors point out the problems of using EtBr staining as a measure
of parasite viability, given the staining of free DNA from dead
parasites. Given that the read out in Figure 2 does not closely match
the microscopic data in Figure 1, the use of this metric seems questionable.
Reply. We have chosen the EtBr-based method of Plasmodium parasite development firstly based on the literature (e.g. Waki et al., Trans R Soc Trop Med Hyg. 1986; Wilson et al., Malar J, 2010; Skorokhod et al., Free Radic Biol Med. 2015). Secondly, in growing parasites, the increase of EtBr signal corresponds the microscopically observed parasite growth and multiplication.
We agree, dead parasites un-expulsed from erythrocytes could be a confounding factor for exact parasitemia calculation by EtBr method. Therefore, to distinguish between vital and dead /damaged parasites we analysed stained smears by manual microscopic inspection and counted the parasitemia of dead and vital parasites separately. These results are reported in Figure 3 (filled column damaged, empty column morphologically normal, even if delayed) . Figure 2 is important to confirm the DHA action at low nM concentrations (1-10nM), clearly decreasing parasite DNA content as compared to DHA-free cultures.
The authors mention in the abstract and the introduction that the
artemisinin class of compounds may have possible antitumor and
immunomodulatory effects in addition to their well-known antimalarial
effects. These possible ‘repurposed’ therapies are not further explored
in the Discussion section. Given the lack of alterations at concentrations where anti-malarial effects take place, perhaps the authors should spend more time discussing these possible non-malarial applications.
Reply. We completely agree with the Referee. We add more information about non-malarial artemisinin applications.
A broader analysis of protein modifications needs to be undertaken prior
to publication. The reader is left questioning whether this
modifications of falcipain-1 are novel to this particular protein (ie
only 1% of parasite proteins are modified) or universal (ie 99% of
parasite proteins are modified).
Reply. Protein modifications by 4-HNE is a not fully understood field. Even under strong exposure to 4-HNE, some specific proteins are heavily modified (similar to scavengers, e.g. serum albumin), some specific proteins are moderately modified (although sometimes with very strong functional consequences, due to modifications in the active site if accessible for 4-HNE), some proteins remain completely un-modified. A number of factors could contribute to 4-HNE modification process: protein composition, folding, binding sites’ accessibility, intracellular compartmentation, anti-oxidative intracellular activity, etc.
In our study several proteins were modified by 4-HNE after DHA action (need to be additionally analysed/confirmed), from which Falcipain-1 is described in this paper. Modified falcipain-1 was not detected in untreated control P.f. cultures analysed at the same time of culturing.
It is unclear what concentration of DHA was used in the mass spectroscopy experiments.
Reply. We performed mass spectrometry analysis under conditions, where 4-HNE-adducts were detectable by FACS/WB and morphological changes in parasites were evident: 1 µM DHA concentration, treatment for 6 hours. Now we report this information in the Results.
It is also unclear how the authors found this specific drug-protein interaction – whether it was via a directed or an unbiased approach.
Reply. The modification of proteins with 4-HNE is not specific for the given drug. Our approach is to study the drug-produced lipoperoxidation under reasonable physiological conditions, with consequent analysis of modified proteins.
Falcipain – 1 is a non-essential protein in P. falciparum; parasites
without this gene function normally in respects to growth and invasion
(PMID: 15166288). It is therefore difficult to imagine that DHA modified
falcipain-1 could explain parasite death. The data presented do not even
comment on the functionality of the enzyme with these modifications. Is
there evidence that it has decreased function when modified in the ways
described?
Reply. We expanded the respective Result paragraph by the analysis of the hypothetical possibility of 4-HNE modifications in the functional sites of i) falcipains 2 and 3 with vital importance for the parasite, ii) human cathepsins, important in the genesis of tumors. Functional consequences of such modifications are discussed now in the paper.
The morphologic and parasite death data that is in the first paragraph
of the Discussion is not novel, and I am not sure should even be covered.
Reply. We reorganized the first part of the Discussion.
Given that ROS and 4-HNE effects are only seen at very high drug
concentration, the physiologic relevance needs to be considered. There
is some mention in Abstract (Line 12) and Intro (Line 61-68) but nothing
in discussion.
Reply. The antitumor and immunomodulatory actions of artemisinin are discussed better now.
This is not a paper about the pharmacologic mechanisms of
Artesunate in parasite infected cells – as these effects do NOT take
place at the nM levels where DHA has its effect. The adduct information
on falcipain-1 is interesting but again needs to be put into context.
Reply. We found 4-HNE modifications in micromolar concentrations which are covering mostly non-antimalarial application of artemisinin. Based on the discovered falcipain-1 modifications we tried to transfer the knowledge to functionally more relevant falcipain 2/3 and to cathepsins, which may be important for oncology studies.
How do the authors hypothesize this to influence parasite death if only seen at uM range?
Reply. We hypothesize the possibility of functional falcipains-2/-3 modifications, due to the similarity of certain 4-HNE binding sites. In any case, we think that 4-HNE-protein interaction is not a principal antimalarial mechanism when nanomolar DHA is applied. It could be an additional mechanism of action of the DHA. An unbiased approach to find 4-HNE-conjugated functionally relevant proteins for P.f. survival at low nanomolar concentrations of DHA is pending to finally decide whether this posttranslational modification is crucially involved in DHA action or not.
Author Response
Major concerns:
The study presented by Skorohod et al, is well designed and interesting bringing new insights into the mechanism of action of DHA against the parasite. The redox process is a tricky phenomenon when dealing with the erythrocytes as hemoglobin, containing the iron atom, can itself become the key player. The authors have shown interestingly that DHA is able to induce ROS production and triggers the lipid peroxidation with the formation of 4-NHE.
Overall, the results are sound. For most of them, DHA already reaches the maximum efficiency at 5 or 10 nM. Increasing the concentrations of DHA did not really enhance the efficiency. How do authors explain this action?
Reply: > DHA is known to kill parasites efficiently at very low doses and the IC50 was determined by several researchers at 1-5 nM for artemisinin-sensitive Plasmodium falciparum strains. Consistently, the drug was efficient in the same low nanomolar range in our synchronized Plasmodium cultures which is the prerequisite for studies on the molecular mechanism of DHA action against parasites.
Fig 1 B shows exemplary most common morphological changes provoked on parasites by short-term exposure (6h) to DHA in a dose dependent manner. Thus, damaged forms of rings after 6 h incubation with 10nM DHA often showed small detached chromatin-containing bodies inside the parasite’s vacuole, distant from the still present purple –coloured ring-structure, and the beginning of chromatin condensation while the overall size of the parasite is widely maintained inside the infected erythrocyte. Even quite normally –shaped rings are visible, as e.g. the upper-left one in the double infected erythrocyte). DHA supplemented for 6 h at 100nM provoked much stronger morphologic alterations with shrunken rings and amorphous, pyknotic parasite forms, which do no longer allow the determination of their developmental stage (Fig 1B).
Parasite growth, assessed by DNA-binding of ethidium bromide (EtBr, Fig 2) at 6 h after DHA treatment shows an exponential decrease up to 10nM DHA and a subsequent linear decrease with a minor negative slope. There are at least two processes that determine the amount of nucleic acids which are quantified by the intercalating EtBr in the in vitro parasite culture 1) the real growth of parasite, i.e. the synthesis rate of nucleic acids. 2) the degradation of nucleic acids in dead parasites and expelling of dead parasites from the host cell. While the first process is efficiently inhibited during the 6 h observation period by DHA at 5 nM and higher, the DNA persists inside the dying parasite and only high doses of DHA (>/=100uM) damage the parasite structure strongly enough during 6 h, that beside the inhibition of DNA replication also the beginning nucleic acid decay may occur. It is clear that 6 h are very short for destructive processes of an intracellularly dying parasite especially in absence of immune cells (as in vitro cultures) and we explain the persistent EtBr signal as the signal of intra-erythrocytic, dead parasites. Further on, after 24 hours, the dose dependent effect is more striking (Fig 3) as total parasitemia for 1000 nM DHA treatment is significantly reduced comparing to the treatment with lower DHA concentrations. Also, the amount of dead parasites in the cultures rises by DHA concentration (Fig. 3).
This idea is summarised in the MS result para 1, p.6:
‘Due to the binding of ethidium bromide to damaged and dying parasites the fluorescence level remained above 0 even at the highest tested DHA concentrations.’
__________________________________________________________________________
In addition, it is important for all the assays to add or explain if the solvents used to prepare the samples (DHA and NAC) can interfere or not. On the figures, the authors should add as control (the complete system with solvent) or state that in the first column, with 0 nM DHA the solvent (medium, distilled water, or Ethanol, DMSO) was used instead of the compound.
Reply: We thank the referee for the important request to improve the method description
We added the requested information in the method paragraph ‘DHA and N-acetylcysteine (NAC) treatment of infected- and non-infected erythrocytes’ . Inserted text in bold. :
‘DHA (Selleck Chemicals GmbH, Planegg, Germany) was dissolved in water-free DMSO to obtain a stock solution of 10mM which was further diluted with DMSO prior to use to obtain DHA-solutions of 1mM, 100µM, 10µM, 5µM and 1µM. The DHA solutions were each added into synchronized parasite culture suspensions atthe final dilution of 1:1000 (to obtain final DHA concentrations of 10 µM, 1 µM, 100 nM, 10nM, 5 nM and 1nM in the parasite-infected erythrocyte suspensions).’ …‘Infected and uninfected erythrocyte control cultures, which were not treated with DHA (0µM DHA), were supplemented with a final concentration of 0.1% (v/v) DMSO, which corresponds to the DMSO concentration of DHA-treated cultures.’
…and radical- and 4-hydroxynonenal (4-HNE)-scavenging N-acetylcysteine (NAC) was dissolved at 100mM in Milli-Q water and added 30 min prior to DHA supplementation into the
We also eliminated ethanol as solvent for DMSO from the MS, as all shown data were obtained with a freshly acquired DHA batch solved in DMSO. Just few pilot experiments aimed to establish best time schedule by light microscopic observation were performed with a DHA –stock solution solved in ethanol, which was immediately available in the lab.
_________________________________________________________________________
DHA-induced cell lysis was excluded by sensitive hemoglobin measurement in the culture suspension (data not shown). For the good understanding of the entire results, the authors should show the results obtained with the luminescence assay.
Reply:>We add Figure 3C: “Absence of DHA-elicited hemolysis.’ Hemolysis was assessed as culture medium heme concentration referred to total haemoglobin-heme concentration in the lysed whole culture ”, Erythrocyte integrity was confirmed for DHA treatment of erythrocytes in 0-10 µM concentration range for 6 h. To exclude schizogony and the expulsion of dead parasites as confounding factors we restricted the observation period for haemolysis to 6 h.
__________________________________________________________________________
In the same way, when dealing with ROS in redox process, cautions must be taken to use a less sensitive ROS probe. During DHA-induced lysis of erythrocytes, there is certainly the release of Hb in the medium or extracellular compartment, resulting in the oxidative process. The ROS production induced by DHA action should be monitored not only by FACS (DCF-DA) but also by using another luminescence probe like luminol analogue (L012) which is less sensitive and able to react with H2O2.
Reply: >We agree with reviewer consideration on extracellular haemoglobin/heme. We checked the absence of erythrocyte lysis in synchronised culture due to 0-1 µM DHA during the observation period, when ROS was determined. .
>We agree with the referee regarding utility of additional ROS test. We are not able to apply immediately L012 probe test, but we have data for the dihydroethdium (DHE) superoxide probe, which was described also to react with hydroperoxide (DHE can be oxidized by H2O2 (Rothe and Valet, J Leukoc Biol. 1990 May;47(5):440-8). We add the Figure 4B with the superoxide test by DHE, confirming the data obtained with DCF probe in Figure 4A.
________________________________________________________________________
I am also wondering as the authors have found the basal production of 4-NHE in non-infected erythrocytes which was not the case for the ROS production assays. This basal production of ROS can also trigger the lipid peroxidation as shown in figure 5 (A, B, C and D). The discrepancies between the two groups can be explained by the specificity of the physiology of each donor in terms of redox state. I would appreciate if the authors took into in consideration this effect to explain the very low ROS production seen in Figure 3.
Reply: RBC as natural oxygen transporters are usually well equipped against ROS explaining low ROS values in RBC of healthy donors. The discrepancy between low ROS and relatively high 4-HNE-
conjugate values in non-infected erythrocytes may result from differences in the half-life of ROS and 4-HNE- conjugates and their assessment. ROS is short living and the assay quantifies active free radicals generated in the moment of the test. Instead, lipoperoxidation product 4-HNE is a stable molecule that binds to proteins forming covalent conjugates that persist and accumulate by time. The relative high levels of 4-HNE-conjugates in Fig 5 measured by conjugate–specific antibodies could be explained by this cumulative effect. Values after 3 hours of DHA action describes three hours of lipoperoxidation chain reaction. Note, non-infected erythrocytes carry also a small basal level of 4-HNE-conjugates originating from eventual pre-experimental oxidative experience in the donor body. The increase of 4-HNE in non-DHA-treated and non-infected RBC by time is mainly due to the transfer of 4-HNE originating from the co-cultured infected erythrocytes (Uyoga et al. Transfer of 4-hydroxynonenal from parasitized to non-parasitized erythrocytes in rosettes. Proposed role in severe malaria anemia. Br J Haematol. 2012 157(1):116-24).
Minor concerns:
On page 12, line 491, read microM and change uM. In the main text, the symbol is already used.
>Corrected
Legend Figure 4, line 363 (read 100 µM NAC were added and not was…).
>Corrected
Overall, even though the results presented in this manuscript seem very interesting, still the authors should give more experimental data particularly for ROS production assay.
>Additional data for ROS production are added in Fig 4.
Major revision is required. Once everything will be completed, the manuscript might be suitable for publication in Antioxidants
Round 2
Reviewer 1 Report
The authors have addressed all of my initial concerns.
There is still need for a thorough editing for English and grammar (ie Line 578 should be exciting, not exiting).
Reviewer 2 Report
The authors have made appropriate modifications and the answers seem fine for me. The revised manuscript can now be accepted for publication in Antioxidants.